# Athermal creep deformation of ultrastable amorphous solids

Pinaki Chaudhuri,[1] Ludovic Berthier,[2] and Misaki Ozawa[3]

[1] *The Institute of Mathematical Sciences, Taramani, Chennai 600113, India*
[2] *Gulliver, UMR CNRS 7083, ESPCI Paris, PSL Research University, 75005 Paris, France*
[3] *Univ. Grenoble Alpes, CNRS, LIPhy, 38000 Grenoble, France*
(Dated: January 31, 2025)

We numerically investigate the athermal creep deformation of amorphous materials having a wide range of stability. The imposed shear stress serves as the control parameter, allowing us to examine the time-dependent transient response through both the macroscopic strain and microscopic observables. Least stable samples exhibit monotonicity in the transient strain rate versus time, while more stable samples display a pronounced non-monotonic S-shaped curve, corresponding to failure by sharp shear band formation. We identify a diverging timescale associated with the fluidization process and extract the corresponding critical exponents. Our results are compared with predictions from existing scaling theories relevant to soft matter systems. The numerical findings for stable, brittle-like materials represent a challenge for theoretical descriptions. We monitor the microscopic initiation of shear bands during creep responses. Our study encompasses creep deformation across a variety of materials ranging from ductile soft matter to brittle metallic and oxide glasses, all within the same numerical framework.

## I. INTRODUCTION

Amorphous solids encompass a wide range of materials, including metallic glasses, colloids, foams, and granular materials. These materials yield (or flow) under external loading in various rheological setups, such as steady-state shearing, shear start-up, and oscillatory shear [1–11].

Amorphous solids like glasses are non-equilibrium materials, with properties that depend for that reason on their preparation history [4]. Notably, the degree of annealing qualitatively alters their mechanical response. Strain-controlled simulations have shown that poorly annealed materials, such as colloids and foams, exhibit ductile yielding, characterized by a continuous stress-strain curve and spatially homogeneous deformation. In contrast, ultrastable materials like metallic glasses, display brittle yielding with a discontinuous stress-strain curve and shear localization [12, 13]. The effects of annealing also significantly influence the rheological behavior in cyclic shear deformation protocols [14–16].

The stress-controlled shear start-up protocol allows us to explore another interesting transient response to external loading, viz. creep deformation. This setup has been extensively studied under both thermal [17–20] and athermal [21–24] conditions through experiments, molecular simulations [25, 26], and coarse-grained models [24, 27, 28]. When the applied stress $\sigma$ is below the yield threshold $\sigma_c$, the strain, $\gamma(t)$, initially increases but eventually reaches a finite plateau value at long times. Hence the strain rate $\dot{\gamma}(t)$ grows initially but then decreases and eventually vanishes when the systems is dynamically arrested. Conversely, when $\sigma$ exceeds $\sigma_c$, the system continues to deform, eventually reaching a steady state flow regime where $\gamma(t) \sim t$ and $\dot{\gamma}(t)$ becomes constant. This constant strain rate for $\sigma > \sigma_c$ corresponds to the steady-state flow curve, usually described by the empirical Herschel–Bulkley (HB) law: $\sigma - \sigma_c \sim \dot{\gamma}^n$, where $n$ is the HB exponent. This steady-state behavior, particularly in the context of the Herschel–Bulkley law, has also been the focus of extensive research [3, 29, 30].

The response to the applied stress exhibits non-trivial signatures at intermediate timescales when $\sigma$ is near $\sigma_c$. Specifically, $\gamma(t)$ increases very slowly over time, exhibiting sub-linear behavior known as creep. As a result, the time evolution of $\dot{\gamma}(t)$ shows a complex shape, which varies depending on the material and its preparation history [26]. Notably, as the yield threshold is approached, an extended creep response with $\dot{\gamma}(t) \sim t^{-\nu}$ (where $\nu$ is an exponent) is observed, which is followed by either the complete cessation of flow, with $\dot{\gamma}(t) \to 0$ when $\sigma < \sigma_c$, or a sudden increase toward fluidization, forming an S-shaped curve with $\dot{\gamma}(t) \sim$ const. when $\sigma > \sigma_c$. The threshold stress (denoted as $\sigma_c$) depends on the initial conditions. More stable materials generally have a higher $\sigma_c$, which is picked up as the height of the stress overshoot in quasistatic strain-controlled analysis [12]. This static yield stress $\sigma_c$ is often distinct from the critical threshold in the Herschel–Bulkley law, which is sometimes referred to as the dynamic yield stress [22, 31, 32], and both stresses may coincide only for poorly annealed samples.

The slow creep deformation process leading to fluidization is characterized by an associated timescale for the onset of flow, known as the fluidization time, $\tau_f$. Specifically, $\tau_f$ diverges as the yield threshold $\sigma_c$ is approached from above, following the relation $\tau_f \sim (\sigma - \sigma_c)^{-\beta}$ [17, 19, 25], where $\beta$ is an exponent which may also depend on the initial stability of the sample [27, 33]. In shear-rate controlled rheology, the fluidization timescale similarly diverges as the shear rate decreases, following $\tau_f \sim \dot{\gamma}^{-\alpha}$, where the exponent $\alpha$ depends on the annealing history of the sample [34]. Recent theoretical work has sought to connect the physics of transient responses (characterized by parameters such as $\nu$, $\tau_f$, $\beta$, and $\alpha$) with steady-state behavior, such as the Herschel–Bulkley law and its exponent $n$ [35, 36]. However, these theo-

retical frameworks have primarily been applied to and validated using experimental and simulation data from soft-matter materials, which tend to fall into the class of less stable systems.

The stability of a sample also affects the spatial manifestation of the fluidization process [26]. Poorly annealed materials tend to exhibit homogeneous fluidization after the transient creep regime, while more stable materials often show sharper shear localization in the regime where $\dot{\gamma}(t)$ increases after reaching a minimum [17, 26]. Theoretically, it has been suggested that this upturn in $\dot{\gamma}(t)$ corresponds to the onset of shear banding [37]. More recently, precursors to fluidization have been observed in experiments during the decreasing branch of $\dot{\gamma}(t)$, prior to large-scale flow onset [38]. However, identifying such precursors from real-space (microscopic) images remains challenging. It is expected that defects or weak spots play a more significant role in stable materials, acting as precursors that lead to the formation of sharper system-spanning shear bands [39–41]. For very stable systems, however, such defects may become extremely rare, and their experimental observation may be complicated, while numerical simulations may even totally miss them [41].

All these questions can be addressed using molecular simulations as they provide microscopic insights into deformation processes. However, standard molecular simulations can only vary the stability of a sample within a narrow range due to the limited simulation timescale. Additionally, it has been argued that directly observing precursors, such as defects or shear band embryos, relevant to macroscopic shear band formation is challenging in molecular simulations. This is because the probability of finding such defects is exponentially suppressed with defect size, and the standard simulation system size is much smaller than that of macroscopic experimental samples [39–41]. The numerical recipe to solve this problem is to introduce by hand localized soft regions, or seed, in the numerical samples and compare the emerging physics to samples with no seed.

Here, we investigate the athermal creep deformation of glasses through molecular simulations, varying the initial stability across an extremely wide range of annealing levels using the swap Monte Carlo algorithm for sample preparation [42]. Our work extends the study of creep deformation, which has been predominantly focused on soft matter systems, to more stable systems relevant to metallic and oxide glasses. Our simulations capture both ductile responses, characterized by relatively homogeneous deformation, and brittle responses, featuring strong S-shaped transient behavior accompanied by sharp shear band formation. We characterize the diverging fluidization timescale as the yield threshold is approached and compare our numerical results with predictions from recent scaling theories, highlighting challenges in describing stable materials. We analyse the shear band formation in real space during the onset of fluidization, and introduce a specific procedure to assess the role played by soft

defects in a sample, thus shedding light on how precursors or shear band embryos develop into system-spanning shear bands.

The manuscript is organized as follows. Section II outlines our numerical models and computational methods. In Sec. III, we describe the macroscopic rheological response of the system and the associated timescales. Section IV provides a visual analysis of the onset of flow under various conditions. Finally, we discuss our results and present conclusions in Sec. V.

## II. MODELS AND METHODS

### A. Model systems

We simulate systems of $N$ size polydisperse spherical particles in cubic and square boxes of length $L$ in three (3D) and two (2D) dimensions using periodic boundary conditions. The pair interaction between particles $i$ and $j$ is a soft-core repulsive potential,

$$u(r_{ij}, d_{ij})/\epsilon = \left(\frac{d_{ij}}{r_{ij}}\right)^{12} + c_0 + c_2\left(\frac{r_{ij}}{d_{ij}}\right)^2 + c_4\left(\frac{r_{ij}}{d_{ij}}\right)^4,$$

$$d_{ij} = \frac{d_i + d_j}{2}\left(1 - 0.2|d_i - d_j|\right),$$

where $r_{ij}$ is the distance between particles $i$ and $j$, $d_i$ is the diameter of the particle $i$, and $\epsilon$ is the energy scale of the potential. The set of parameters, $c_0$, $c_2$, and $c_4$, are adjusted so that the potential and its first and second derivatives vanish at the cutoff distance $r_{\text{cut},ij} = 1.25d_{ij}$. The particle diameters are drawn randomly from a continuous size distribution $P(d) = A/d^3$ in the range $[d_{\min}, d_{\max}]$, where $A$ is normalizing constant. We use parameters such that $d_{\min}/d_{\max} = 0.45$ and the average size diameter is $\bar{d} = 1.0$ and sets the unit length. These two models have been carefully studied before [42–45]. We perform simulations at constant number density $\rho = 1.02$ for 3D, and $\rho = 1$ for 2D, using $N = 96000$ in 3D and $N = 64000$ in 2D. We report macroscopic observables, such as $\dot{\gamma}(t)$, obtained from averaging over 25 independent samples. We mainly present the data in 3D, yet due to its visual clarity, we show some 2D data for the time evolution of the shear band formation.

### B. Preparation of amorphous states with controlled stability

To prepare glassy samples with different stabilities at temperature $T = 0$, we first equilibrate the system at a finite temperature, $T_{\text{ini}}$, using the efficient swap Monte Carlo method [42]. These equilibrium configurations are then instantaneously quenched to $T = 0$ using the conjugate gradient method [46].

We generate glassy samples with initial temperatures $T_{\text{ini}} \in [0.062, 0.200]$ in 3D, covering a broad range of initial stability. Within this temperature range, we observe

a spectrum of yielding behaviors from brittle to ductile when using strain-controlled athermal quasi-static (AQS) shear simulations [12].

In 2D, the initial preparation temperature is $T_{\mathrm{ini}} = 0.035$, which places the system extremely deep within the glassy regime [43], leading to brittle yielding [13] characteristic of ultrastable systems.

To introduce a soft region, or seed, within an otherwise stable glass, we follow the method developed in Ref. [41]. We define an ellipsoidal region characterized by the major axis length $D_a$ and the minor axis length $D_b$. In this study, we set $D_a = 50$ and $D_b = 8$, which are much smaller than the linear box length of the 2D system $L = 253$. We then perform additional swap Monte Carlo simulations restricted to the particles within the ellipsoidal region, while the particles outside remain pinned. The temperature for these additional Monte Carlo simulations is set to $T_{\mathrm{h}} = 10.0$. The dynamical mode-coupling crossover temperature of the system is $T_{\mathrm{mct}} \approx 0.110$, meaning that $T_{\mathrm{h}}$ is about 100 times higher than this mode-coupling temperature. After these high temperature Monte Carlo steps, we quench the obtained configuration back to zero temperature using the conjugate gradient method. As a result of this protocol, the final glass samples contain a poorly-annealed ellipsoidal seed region immersed in a much large ultrastable glass matrix.

### C. Numerical method for mechanical loading

The response to applied shear stress is studied using molecular dynamics (MD) simulations, following the method described in Ref. [25]. This approach involves integrating the equations of motion for the constituent particles, as well as the equation governing the macroscopic strain rate, $\dot{\gamma}(t)$, which emerges from the imposed shear stress, $\sigma$. The shear rate must adjust itself in order to maintain a constant applied stress, and this is ensured using a feedback control scheme [25, 47]:

$$\frac{\mathrm{d}\dot{\gamma}(t)}{\mathrm{d}t} = B\left[\sigma - \sigma_{xy}(t)\right], \qquad (1)$$

where $\sigma_{xy}$ is the Irving-Kirkwood expression for the instantaneous shear stress, including the kinetic contribution, and $\sigma$ the desired value of the applied stress. The damping parameter is set to $B = 1$. Additionally, to control dissipation under athermal conditions, we apply a Langevin thermostat that couples only to the $y$-component of the particle velocities [32]; the corresponding damping timescale is set to 0.1.

## III. MACROSCOPIC RHEOLOGICAL RESPONSE

### A. Creep responses and flow curves

We begin by investigating the macroscopic rheological responses through flow curves. In Figs. 1(a, b, c), we present the mean strain rate versus time curves, $\dot{\gamma}(t)$, for the three-dimensional model, averaged over 25 independent samples. These data feature three representative stability levels of the initial samples [12]: poorly annealed glasses (Fig. 1(a)), where no stress overshoot is observed in the stress versus strain curve; modestly annealed glasses (Fig. 1(b)), which exhibit a mild stress overshoot; and ultrastable glasses (Fig. 1(c)), characterized by a large stress overshoot and a sharp, discontinuous stress drop. With this choice, the corresponding materials in real experimental systems could be wet foams for poorly annealed glasses, colloidal glasses for modestly annealed glasses, and metallic glasses for ultrastable glasses.

For poorly annealed glasses in Fig. 1(a), when the imposed stress is low ($\sigma = 0.14$), there is a prolonged creep regime, where $\dot{\gamma}(t) \sim t^{-\nu}$ with $\nu = 0.62$, and $\dot{\gamma}(t)$ decays to zero over time, i.e., the system remains in a dynamically arrested state. As the stress increases, particularly for $\sigma \gtrsim 0.18$, $\dot{\gamma}(t)$ reaches a plateau, indicating the onset of steady-state flow. For a narrow range of intermediate stress values, $0.14 < \sigma < 0.18$, a fraction of trajectories within the ensemble remain stuck while others yield and attain steady flow [26].

Determining the exact threshold stress, above which the system flows indefinitely, is thus challenging from the mean strain rate versus time curves alone, as the threshold varies from sample to sample within this ensemble of finite-size systems. Precise determination requires careful simulations for each sample (as discussed later). Therefore, we will use an alternative approach to estimate the threshold stress, as explained below.

We next examine modestly annealed samples, as shown in Fig. 1(b). For these samples, we observe a power-law time decay of $\dot{\gamma}(t)$, with eventual termination for $\sigma = 0.18$. This indicates that while $\sigma = 0.18$ is sufficient to induce steady-state flow in poorly annealed glasses (Fig.1(a)), this value is no longer sufficient for modestly annealed samples, thus qualitatively demonstrating the dependence of the threshold yield stress on the degree of annealing. Furthermore, the creep exponent $\nu = 1.25$ is larger than in the poorly annealed samples, indicating another aspect of the stability dependence. When the imposed stress is increased ($\sigma \gtrsim 0.22$), $\dot{\gamma}(t)$ exhibits a near power-law decay (with a different exponent) at intermediate timescales. At longer times, $\dot{\gamma}(t)$ increases and enters the steady state, resulting again in an S-shaped curve.

We now turn to ultrastable glasses, where even under a relatively high stress of $\sigma = 0.41$, the mean $\dot{\gamma}(t)$ exhibits a power-law decay to zero. Additionally, we find that the exponent $\nu$ increases further, reaching $\nu = 1.75$. When

samples and the ensemble average broadens the transition in finite systems. The significantly enhanced threshold stress, which lies between 0.41 and 0.44, is consistent with the stress overshoot values obtained in strain-controlled AQS simulations of the same system [12].

The strong and systematic dependence of the threshold stress $\sigma_c$ and of the exponent $\nu$ on the degree of annealing is a key finding of this section.

The final plateau value of the strain rate, $\dot{\gamma}_{ss}$, depends on the imposed stress but not on the initial sample stability, resulting in the steady-state flow curve shown in Fig. 1(d). We fit this data using the Herschel-Bulkley function, $\sigma = \sigma_d + A\dot{\gamma}^n$, and estimate the dynamical yield threshold to be $\sigma_d = 0.144$ and the HB exponent to be $n = 0.41$. These values are consistent with those reported from AQS and finite strain rate simulations for the same model [12, 48]. Note that the steady-state strain rate values do not depend upon preparation histories, since the memory of the initial stability is completely erased after entering the steady-state flowing regime.

## B. Timescales and thresholds

One key feature of the flow curves in Figs. 1(a, b, c) is that the time needed to reach steady-state flow for $\sigma > \sigma_c$ strongly depends on the applied stress. Notably, this timescale appears to diverge as we approach the threshold stress from above. To quantify this, we define a fluidization timescale, $\tau_{ss}$, which quantifies the time it takes for the mean shear rate $\dot{\gamma}(t)$ to reach its steady-state value. In Fig. 2(a), we show how $\tau_{ss}$ varies with the imposed stress, $\sigma$, across the different initial stabilities we studied. For poorly annealed glasses ($T_{\mathrm{ini}} = 0.200$), $\tau_{ss}$ increases as $\sigma$ decreases and seems to diverge at a finite stress (or threshold stress). As the stability increases, $\tau_{ss}$ appears to diverge at higher stresses, consistent with the larger thresholds for more annealed glasses seen in Figs. 1 (b, c).

We then apply a widely used empirical fitting function, $\tau_{ss} \sim (\sigma - \sigma_c)^{-\beta}$, shown by dashed curves in Fig. 2(a). This function helps us estimate the mean yield threshold $\sigma_c$ and the divergence exponent $\beta$, listed in Table III C. As expected we observe a systematic variation in $\sigma_c$ from 0.148 (for $T_{\mathrm{ini}} = 0.200$) to 0.425 (for $T_{\mathrm{ini}} = 0.062$), which represent a significant increase by a factor of about 3. These values are quite similar to the evolution of the height of the stress overshoot obtained in AQS simulations for these initial stabilities, as reported in Ref. [12]. We observe that the divergence exponent $\beta$ also varies considerably with stability, ranging from 1.22 (for $T_{\mathrm{ini}} = 0.062$) to 2.04 (for $T_{\mathrm{ini}} = 0.200$). This trend is consistent with findings from studies on mesoscopic elastoplastic models [27, 33].

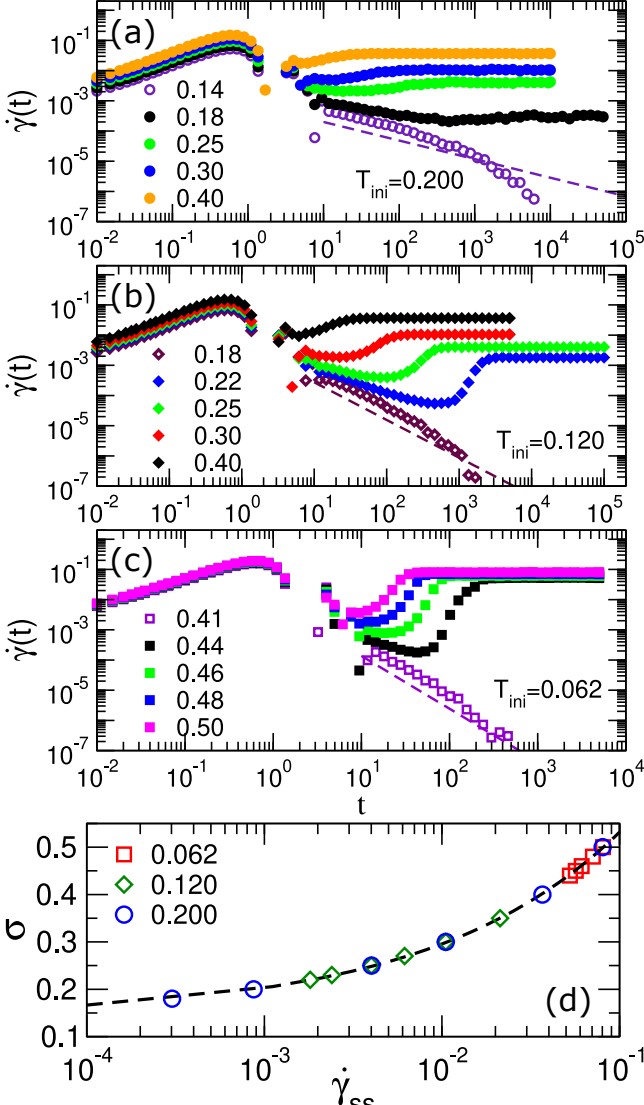

FIG. 1. (a, b, c): Time evolution of ensemble-averaged shear-rate in response to imposed stresses $\sigma$ (as marked) for three-dimensional initial amorphous states prepared via quench from $T_{\mathrm{ini}} = 0.200$ (a), $T_{\mathrm{ini}} = 0.120$ (b), and $T_{\mathrm{ini}} = 0.062$ (c). Filled symbols correspond to cases where steady-state flow is observed in all samples, whereas open symbols are used for cases where the system remains solid in all samples. Dashed lines correspond to $\dot{\gamma}(t) \sim t^{-\nu}$ with $\nu = 0.62$ (a), 1.25 (b), and 1.75 (c). (d): Steady-state flow curve, viz. the evolution of the imposed stress with the steady-state ensemble-averaged shear-rate obtained by gathering data for $t \to \infty$ in (a, b, c), for flowing states. The dashed curve corresponds to the Herschel-Bulkley law, with $\sigma_d = 0.144$ and $n = 0.41$.

a much larger stress is applied ($\sigma \gtrsim 0.44$), $\dot{\gamma}(t)$ shows a more pronounced S-shaped curve, eventually reaching steady-state flow. Note that individual samples display sharp, discontinuous jumps in $\dot{\gamma}(t)$ after the intermediate creep decay (as shown in other figures below). However, the mean curve in Fig.1(c) appears smoother because the discontinuous jumps occur at different times in different

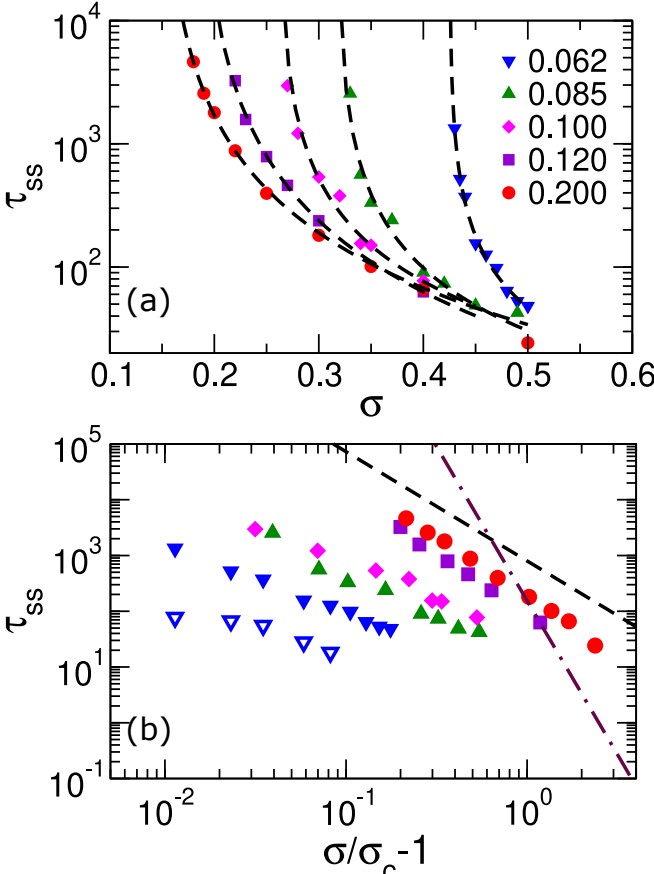

| $T_{ini}$ | $\sigma_c$ | $\beta$ |
|---|---|---|
| 0.062 | 0.425 | 1.22 |
| 0.085 | 0.317 | 1.47 |
| 0.100 | 0.261 | 1.49 |
| 0.120 | 0.183 | 2.15 |
| 0.200 | 0.148 | 2.04 |

TABLE I. List of estimated yield threshold, $\sigma_c$, and exponent $\beta$ for the different preparation histories, determined from the divergence of the fluidisation time.

seen in Figs. 1(a, b, c). These values may align with the theoretical predictions, but we lack the numerical resolution to clearly separate the two scaling regimes, $\sigma < \sigma_c$ and $\sigma = \sigma_c$. This is because it is difficult to observe a power-law regime far below $\sigma_c$ in our finite-size simulations ($N = 96000$), and also hard to very precisely pinpoint $\sigma = \sigma_c$ due to significant sample-to-sample fluctuations mentioned above. Dedicated computational work is needed to resolve this issue and improve the comparison with theory, presumably using much larger system sizes to enhance the time window over which power law decay can be observed.

FIG. 2. (a): Fluidization timescale, $\tau_{ss}$, as a function of applied shear stress, $\sigma$, for states having different $T_{ini}$. Dashed curves correspond to a power law fitting, $\tau_{ss} \sim 1/(\sigma - \sigma_c)^\beta$ with $\sigma_c$ and $\beta$ are listed in Table III C. (b): Same data shown as a function of $(\sigma/\sigma_c - 1)$, compared to theoretical predictions $\beta = 1/n - 1/2$ in Ref. [36] (dashed line) and $\beta = 9/(4n)$ in Ref. [35] (dotted-dashed line), using our numerical estimate of $n$ in Fig. 1(d). Open symbols show estimates for the timescale $\tau_m$, defined via the minimum of $\dot{\gamma}(t)$, for $T_{ini} = 0.062$.

### C. Comparisons with scaling theories

Next we discuss our findings in the context of some recent theoretical scaling predictions which we presented in the introduction.

Popović *et al.* [36] have developed a scaling theory by extending the steady-state Herschel-Bulkley law to a time-dependent transient regime, based on the underlying stress versus strain curve. The theory describes power-law behaviors below, at, and above the threshold stress $\sigma_c$. For $\sigma < \sigma_c$, it predicts creep decay, $\dot{\gamma}(t) \sim t^{-\nu}$, with $\nu = 1/(1-n)$. In our case this would gives $\nu \simeq 1.69$, using our numerically determined HB exponent $n = 0.41$. At $\sigma = \sigma_c$, theory predicts $\nu = 2/(2-n) \simeq 1.26$ for systems with a stress overshoot, while $\nu = 1$ is predicted for systems without a stress overshoot, such as poorly annealed glasses. Our simulations show $\nu = 0.62$, $1.25$, and $1.75$ for $T_{ini} = 0.200$, $0.120$, and $0.062$, respectively, as

For $\sigma > \sigma_c$, the system eventually fluidizes, and the theory predicts $\tau_{ss} \sim (\sigma - \sigma_c)^{-\beta}$ with $\beta = 1/n - 1/2 \simeq 1.96$. This value is comparable to our simulations for poorly annealed glasses ($T_{ini} = 0.200$) and slightly annealed glasses ($T_{ini} = 0.120$). However, our data for more stable glasses deviates systematically from 1.96 as stability increases (or $T_{ini}$ decreases). This suggests a challenge for scaling theories describing the creep response of stable glasses, like metallic glasses, which should account for stability. We also note that in Popović *et al.* [36] the fluidization timescale is defined by the minimum in the time evolution of the averaged $\dot{\gamma}(t)$, rather than the time when steady state is reached. However, even if we use the fluidization timescale as the minimum in $\dot{\gamma}(t)$, called $\tau_m$, the trend is similar to our estimated $\tau_{ss}$ (as shown by open symbols in Fig. 2(b)). This coincidence between the two timescales is consistent with the empirical Monkman-Grant relationship that reports a linear correlation between these two timescales [49, 50].

In a separate study, Benzi *et al.* [35] studied the time evolution of a fluidity model which includes a gradient term to take care of non-local effects. They measure the timescale for shear band formation in soft materials like dense emulsions, colloidal gels, microgels, and foams, and the model is shown to predict $\tau_{ss} \sim (\sigma - \sigma_d)^{-\beta}$ with $\beta = 9/(4n) \simeq 5.53$. As shown in Fig. 2(b), this value of the exponent does not adequately capture the divergence observed in our data, even in the case of the poorly annealed state where $\sigma_c$ is close to $\sigma_d$.

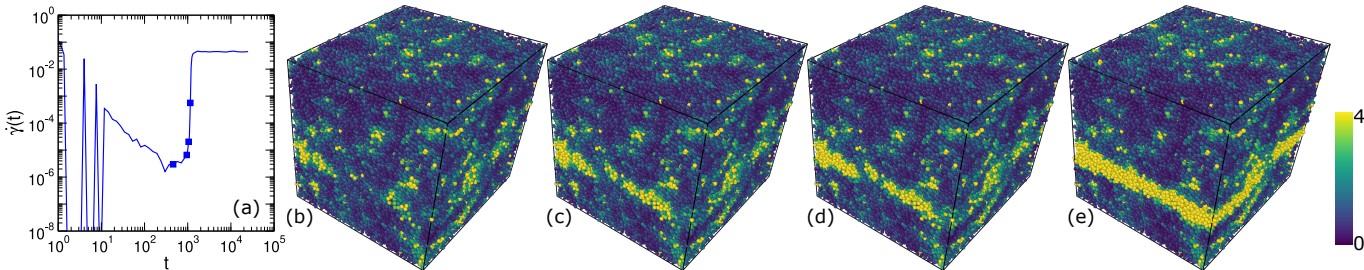

FIG. 3. (a): Time evolution of the strain rate for a 3D ultrastable glass sample ($T_{\mathrm{ini}} = 0.062$) under constant imposed stress $\sigma = 0.42$, i.e. just above $\sigma_c$ corresponding to this initial state. (b-e): Maps of accumulated plastic events measured via non-affine displacements $D^2_{\min}$, at $t =$459.35 (b), 950 (c), 1050 (d), and 1150 (e), marked as square points in (a).

## IV. SPATIAL ANALYSIS OF ONSET OF FLOW

After analyzing the global macroscopic response in the previous Sec. III, we now focus on how the response to applied shear stress is spatially organized. Specifically, our objective is to analyze the spatial signatures characterizing the onset of flow. Previous studies have shown that in well-annealed states, large-scale flow emerges through the intermediate formation of shear bands, i.e., spatially heterogeneous dynamic structures [12]. This contrasts with poorly annealed states, where the onset process is spatially much more homogeneous [51].

Here, we primarily investigate the precursors to shear band formation in ultrastable glasses (for poorly-annealed glasses and slightly annealed glasses, see earlier studies Ref. [26]). To study the emergence of such spatially heterogeneous flow structures, we create spatial maps of local plasticity, measured relative to the undeformed amorphous state at $t = 0$. The local plasticity at any given time is quantified by measuring the non-affine displacements of each particle, $D^2_{\min}$, between its position at time $t$ and its initial position before shear stress is applied, i.e. $t = 0$. This measure reflects the local deviation of particle displacement from affine deformation and has been widely used to characterize local plastic responses [5].

### A. Visualizing failure in three dimensions

In Fig. 3, we present the analysis for an ultrastable 3D glass sample quenched from $T_{\mathrm{ini}} = 0.062$. The evolution of the strain rate $\dot{\gamma}(t)$, for the trajectory of this initial state, during the onset of flow is shown in Fig. 3(a) for the smallest stress ($\sigma = 0.42 > \sigma_c$), where steady flow is observed at long times. The strain rate exhibits a prolonged power-law decay followed by a sudden, nearly discontinuous jump toward fluidization. The snapshots in Figs. 3(b-e) show a sequence of maps of the local $D^2_{\min}$. The corresponding time frames for these spatial maps are also marked in Fig. 3(a).

The earliest time point where we can visualize the first precursor of what will eventually become a shear band is at $t = 459.35$, just after the minimum in $\dot{\gamma}(t)$, where the macroscopic strain rate begins increasing after the initial decrease (see Fig. 3(b)). This increase can be attributed to the propagation of locally yielded spots. Over time, more damaged spots become visible in the same plane, which gradually merge to eventually form a shear band, as seen in Figs. 3(c-e). This merging and growth process occurs rapidly, as indicated by the sharp increase in strain rate (see Fig. 3(a)) in an avalanche-like manner between these frames. While other locally yielded spots appear within this resolution of $D^2_{\min}$, the flow process only accelerates when the relevant spots propagate, leading to the emergence of the macroscopic shear band, which eventually broadens to fluidize the entire system at much larger times.

### B. Visualizing failure in two dimensions

For better visualization and to access larger linear sizes of the simulation domains, we now switch our analysis to 2D systems. In this analysis, we focus again on ultrastable amorphous states, specifically sampled from $T_{\mathrm{ini}} = 0.035$.

In Figs. 4(b-h), we present a sequence of maps of the local $D^2_{\min}$ computed during the trajectory of an initial state sampled from the $T_{\mathrm{ini}} = 0.035$ ensemble, for the smallest stress $\sigma = 0.43880 > \sigma_c$ at which steady flow is observed at long times. The corresponding time evolution of the macroscopic strain rate $\dot{\gamma}(t)$, measured for this trajectory, is shown in Fig. 4(a), where the onset of flow is sharply marked by a rapid increase around $t \approx 10^5$, after a prolonged decrease in the strain rate, similarly to the 3D system.

The maps reveal that, early on at $t = 800$, a vertical locally yielded spot appears (see Fig. 4(b)). However, this spot does not contribute to the eventual shear band formation. Much later, at $t = 169000$ (Fig. 4(d)), another vertical yielded spot emerges, which acts as the source of the subsequent cascade of plasticity. The horizontal spots that appear later, at $t = 169650$ (Fig. 4(e)), result from this cascade and are connected via periodic boundary conditions. These horizontal spots spread and merge to form a network of plastic events, eventually de-

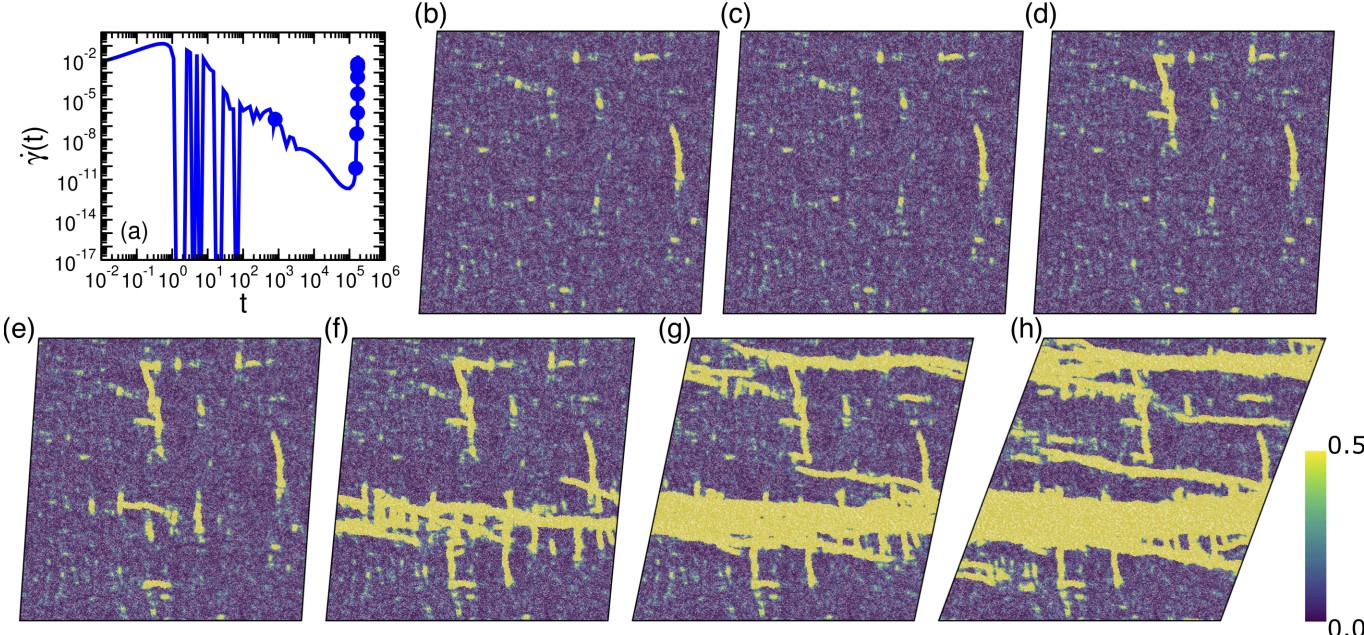

FIG. 4. (a): Time evolution of the strain rate for a 2D ultrastable glass sample ($T_{\mathrm{ini}} = 0.035$) under constant imposed stress $\sigma = 0.43880$, i.e., just above $\sigma_c$ corresponding to this initial state. (b-h): Maps of accumulated plastic events measured via non-affine displacements $D^2_{\min}$ at $t = 800$ (b), 160000 (c), 169000 (d), 169650 (e), 169800 (f), 169900 (g), and 169950 (h), marked as points in (a). These maps demonstrates how yielding proceeds via the relevant precursors.

veloping into a system-spanning horizontal shear band at $t = 169800$ (Fig. 4(f)). A second shear band then forms at the top, which connects with the bottom shear band to drive the system into steady flow (Figs. 4(g) and (h)). The emergence of the second shear band is reasonable since the typical distance between shear bands, $\xi$, scales with the strain rate as $\xi \sim \dot{\gamma}^{-a}$, where $a > 0$ is an exponent [48]. This distance may become smaller than the linear box length of the simulation when the strain rate is large.

All of these events occur within a relatively short time window, as only a time $\Delta t = 900$ separates frames (d) and (h), to be compared to the much longer time $t \approx 10^5$ spent since the stress was first applied. This rapid sequence of catastrophic events highlights the avalanche-like nature of the cascading plasticity that leads to the eventual failure and flow.

## C. Bifurcation near critical stress

Motivated by a previous study [26], we now perform a bifurcation analysis between the arrested state ($\sigma < \sigma_c$) and the flowing state ($\sigma > \sigma_c$), varying $\sigma$ by a very small amount around the value of $\sigma_c$ that characterizes a given sample. We examine the same initial state discussed above in Fig. 4, comparing the response to two slightly different applied shear stresses with a difference of $\delta\sigma = 10^{-5}$, leading to a relative stress difference $\delta\sigma/\sigma_c \approx 2.2 \times 10^{-5}$. The results are shown in Fig. 5.

The time evolution of the macroscopic strain and strain rate is shown in Figs. 5(a) and (b) for $\sigma = 0.43879$ (arrested state) and 0.43880 (flowing state), respectively. For the smaller stress, the dynamics become arrested at long times, indicated by the asymptotic vanishing of the strain rate. When the applied stress is only slightly larger ($\delta\sigma = 10^{-5}$), the system initially follows the same trend as with the smaller stress until around $t \approx 3.4 \times 10^4$, when a non-monotonic behavior sets in and $\dot{\gamma}(t)$ abruptly increases, signaling the rapid onset of flow, as discussed in detail in Fig. 4.

The snapshots in Figs. 5(c-f) show the corresponding maps of local $D^2_{\min}$ in both cases. Up to $t = 34280$, near where the two trajectories bifurcate in the $\gamma(t)$ and $\dot{\gamma}(t)$ curves in Figs. 5(a, b), the spatial response is virtually identical, with some yielded spots appearing in both trajectories. However, as mentioned earlier, these spots do not contribute to the eventual onset of flow. A second yielded spot appears at a later time for the higher stress but not for the smaller stress, and this alone determines the final outcome of the two trajectories, either asymptotic arrest or flow. This analysis highlights that for the eventual failure process to kick in, a single significant soft spot may be able to eventually nucleate a macroscopic shear band. Given how close the two trajectories are up to the minimum of the shear rate, it may appear vain to try and predict from an early precursor analysis the eventual macroscopic failure of a particular material in a particular trajectory.

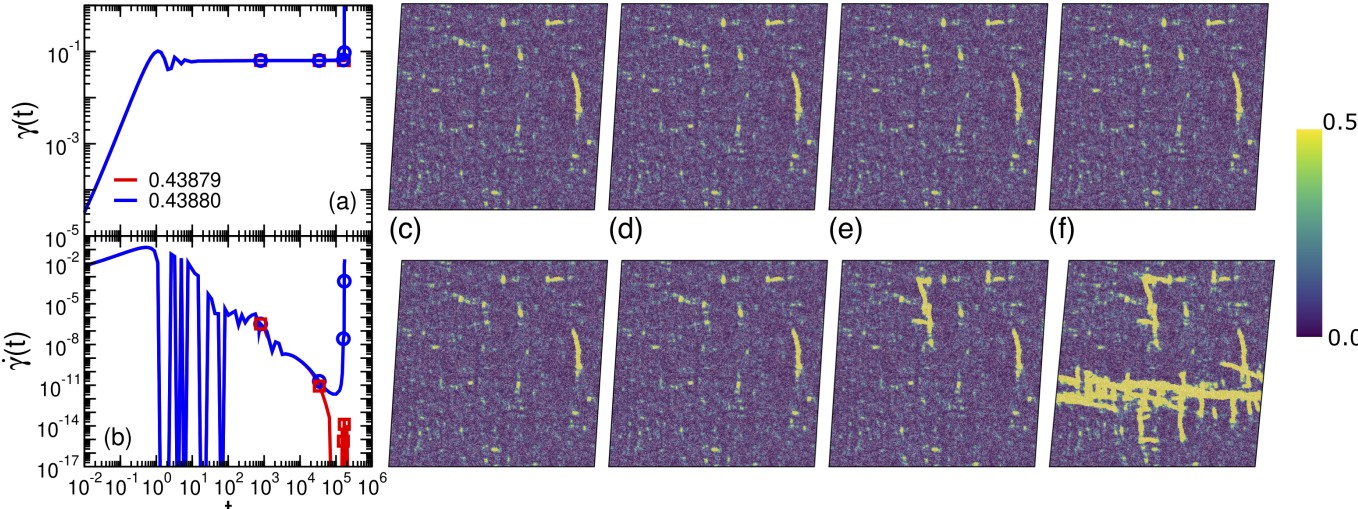

FIG. 5. Comparative response of the same initial state, analyzed in Fig. 4, to imposed stresses of $\sigma = 0.43879$ and $0.43880$, i.e., a difference of $10^{-5}$. (a, b): Time evolution of observed strain (a) and strain-rate (b) responses. (c-f): Sequence of maps of non-affine displacements $D^2_{\min}$, at the points marked in (a, b), viz. $t =800$ (c), $34280$ (d), $160000$ (e), and $169800$ (f), for $\sigma = 0.43879$ (top) and $\sigma = 0.43880$ (bottom).

## D. Failure in solid with soft seed

The spatio-temporal studies in Figs. 3 and 4 have shown that the early soft spots that show up during the response to an applied stress may not necessarily act as precursors to the eventual shear band. In the case of strain-controlled yielding, it has been argued that in macroscopic samples, rare defects (presumably larger than those observed in Figs. 3 and 4) trigger the formation of the macroscopic shear band. As these defects are exponentially rare in size, they are virtually impossible to observe in molecular simulations of limited sizes, thus leading to severe finite size effects in molecular simulations compared to experiments that cannot simply be handled by increasing the linear size of the system.

To address this limitation, recent simulations have introduced such a defect (or seed) manually into the simulation box. As a result, the effect of rare seeds can be numerically analyzed without increasing exponentially the linear size of the system. In practice, in Ref. [41], a soft ellipsoidal region was seeded into a stable amorphous solid, and the subsequent mechanical response, probed via athermal quasistatic shear, confirmed that the seeded weak region acted as the embryo of the emerging shear band, leading to the failure of the solid. Furthermore, it was shown that this localised seeding region could reduce the height of the global stress overshoot, implying that the static yield stress of the solid decreased due to the presence of a single localised defect [40]. We now investigate how the presence of a similar seed influences the response to an applied shear stress.

In Fig. 6, we provide an extensive account of our analysis. Note that, for studying the response to seeding, we use the same initial state discussed in Fig. 4 (where no seed is employed), but now with a soft seed placed at the center. Similar to our analysis in Fig. 5, we determine the lowest shear stress ($\sigma_0 = 0.4012$) at which steady flow is observed at long times, and the largest applied stress ($\sigma_0 = 0.4010$) where no large-scale failure occurs and the system arrests. These values need to be compared with $\sigma_c \approx 0.4388$ when no seed is present.

The corresponding maps of local $D^2_{\min}$ are shown in Figs. 6(C) and (D), while the comparative time evolution of shear strain and shear rate for these two cases is presented in Figs. 6(A). The key point to note is that, even though plasticity is observed in the weak zone for $\sigma = 0.4010$, it does not affect the surrounding area. Only when the stress is slightly increased to $\sigma = 0.4012$ does the shear band initiate, leading to a cascade towards failure, as shown in Fig. 6(D). Therefore, we conclude that the presence of a weak zone alone is not sufficient to cause the material to yield and a sufficient amount of stress must be applied to trigger plasticity in the zones adjacent to the weak spot, leading to shear band formation. However, the situation changes in the regime $0.4012 \lesssim \sigma \lesssim 0.4388$, where the seeded solid eventually fails while the pristine glass does not. (This is shown in 6(B) where the original glass sheared at $\sigma_0 = 0.4012$ shows very little plasticity.) In this regime therefore, the plasticity observed in the soft seed serves to nucleate the macroscopic shear band. This directly demonstrates how a localized soft region can depress the macroscopic yield stress of the material.

Overall, this analysis demonstrates that the manually inserted localised seed, which would spontaneously appear in a real macroscopic sample, lowers the macroscopic threshold yield stress and triggers eventual failure by forming a shear band, provided the applied stress is

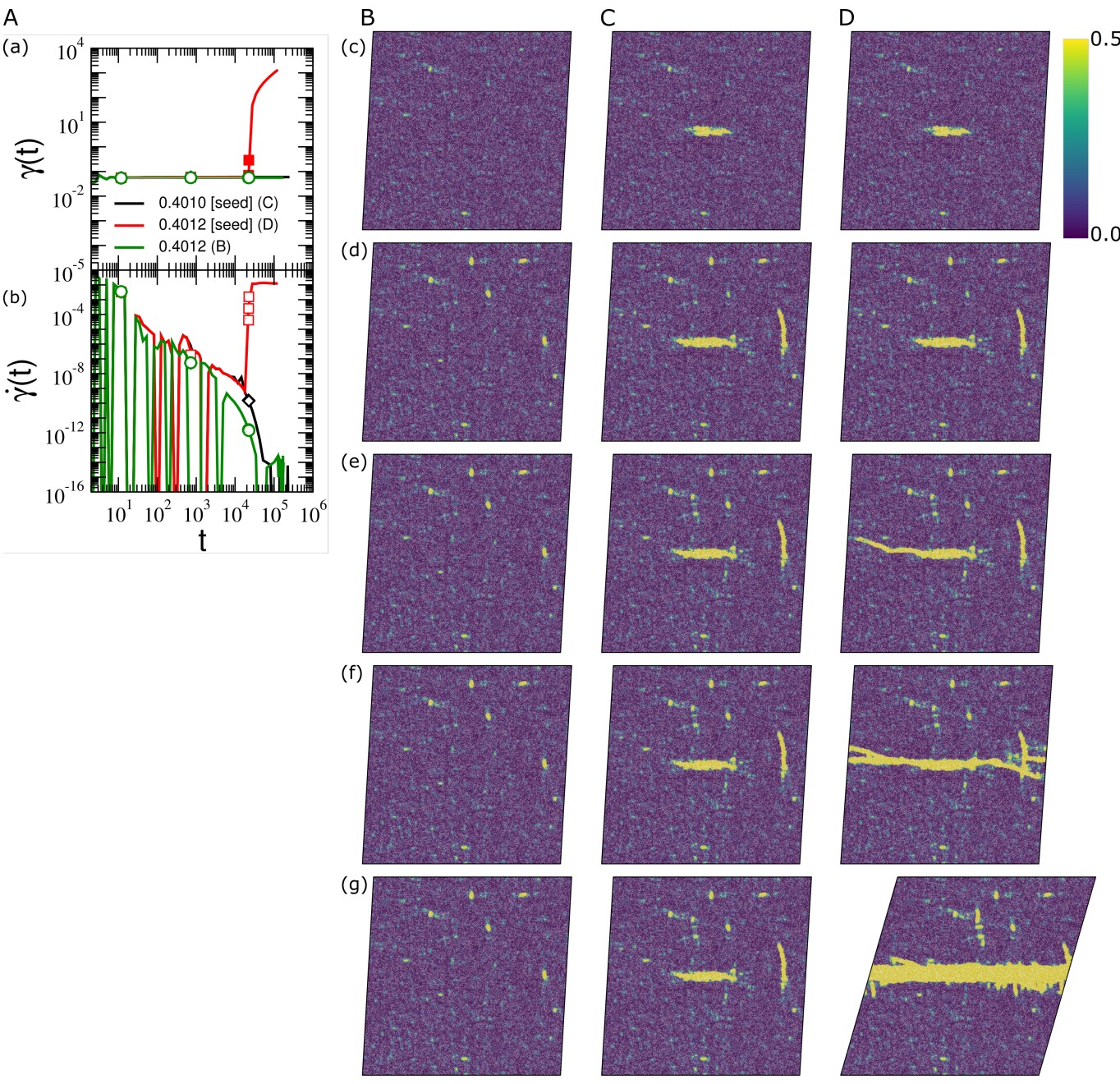

FIG. 6. Analysis of the influence of a soft seed in ultrastable glass. (A): Time evolution of the observed strain (a) and strain-rate (b) response for both seeded and non-seeded states. (B, C, D): Sequence of maps of non-affine displacements, $D_{\min}^2$ at the marked points for imposed stresses of $\sigma = 0.4010$ (Column C) and $\sigma = 0.4012$ (Columns B and D). The maps correspond to times $t = 11.75$ (c), 725 (d), 22150 (e), 22250 (f), and 22500 (g), as indicated in (A).

sufficient enough. This conclusion is the counterpart, for creep flows, of the previous analysis performed in strain-controlled AQS studies [41].

## V. CONCLUSION AND DISCUSSION

We performed molecular simulations of athermal amorphous solids under constant stress, varying the initial sta-

bility significantly using the swap Monte Carlo algorithm, ranging from poorly annealed to ultrastable glasses. This comprehensive study monitored both macroscopic (strain rate flow curves, fluidization timescales) and microscopic (spatial maps of flow onset) observables. Our results show that creep responses and fluidization processes strongly depend on preparation history both qualitatively and quantitatively. Poorly annealed glasses exhibit a gradual evolution of the strain rate, while ultra-

stable glasses display sudden, discontinuous-like jumps in the strain rate, associated with a sharp system-spanning shear band after prolonged creep decay. We also computed fluidization timescales, which diverge near the yield stress, whose strength dependence on glass stability. The associated power laws and exponents were extracted and compared with recent scaling theory predictions. Lastly, we investigated the fluidization mechanism in ultrastable glasses in real space, revealing that a weak spot, or shear band precursor, inserted in the sample grows into a system-spanning shear band and modifies the macroscopic onset of flow.

Our study covers the creep responses of materials across a wide range of stabilities, including poorly annealed glasses like foams and emulsions, slightly annealed glasses like colloids, and ultrastable glasses like metallic and oxide glasses. Our numerical data for less stable glasses show some reasonable agreement with the recent scaling theory by Popović *et al.* [36] although direct quantitative tests would require dedicated studies including much larger systems. However, for more stable glasses, clear discrepancies arise between available theoretical predictions and our data. This highlights the need for a scaling theory that accounts for the material's stability more explicitly and can treat brittle materials.

In this paper, we impose a sudden constant stress and observe the time evolution of strain and strain rate as part of the creep deformation process. Similar phenomena can be seen in cases where strain is suddenly imposed and the stress response is monitored, leading to a delayed timescale for material failure [52]. Related physics can also be observed in fatigue failure under cyclic deformation, where the number of cycles prior to failure depends on factors such as the degree of annealing [53, 54]. It would be interesting to discuss the timescales for transient responses across different deformation settings from a unified perspective.

## ACKNOWLEDGMENTS

We thank K. Martens, M. Popović, A. Rosso, and M. Wyart for discussions. We thank HPC facility at IMSc for computational resources. L.B. acknowledges the financial support of the ANR THEMA AAPG2020 grant.

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
