# Peer review of "Athermal creep deformation of ultrastable amorphous solids"

_SciPost Physics, doi:SciPost Phys. 19, 092 (2025)_

## Round 1 · Referee Report · Anonymous (Referee 1) · 2025-3-29

Strengths

1) The aims of the manuscript are twofold and intersting. 2) the numerical work is careful and at the state of the art

Weaknesses

1) more care to the finite size effect study and sample to sample fluctuations

Report

The manuscript investigates the creep regime at zero temperature in atomistic simulations of amorphous materials with varying degrees of annealing. The authors prepare samples with three different annealing levels and study the strain rate under an applied stress . A threshold stress is identified, $\sigma_c$, which increases for better-annealed materials. For $\sigma<\sigma_c$ , the strain rate exhibits a power-law decay, whereas for $\sigma>\sigma_c$ , the strain rate reaches a minimum before the onset of a fluid shear band.

The aims of the manuscript are twofold: (1) to characterize the threshold , the power-law decay, and the shear band formation time as functions of annealing; (2) to explore the presence of precursors that signal the formation and location of the shear band.

Overall, the manuscript is well written and addresses an interesting problem. However, I have some concerns regarding certain conclusions:

(A) Initial Growth of Strain Rate:
The manuscript does not mention the initial growth of the strain rate. What is the origin of this growth phase? Is it related to an initial elastic response or transient effects?

(B) Power-Law Decay Below :
The authors report that for , the strain rate follows a power-law decay with an exponent expected to be independent of annealing. However, the results indicate a very low exponent in poorly annealed samples, even below unity. This cannot be the long-time decay discussed in Popović et al., since the total strain is the time integral of the strain rate and would diverge for . From Figures 1a and 1b, it seems that at long times, the decay becomes faster. It would be interesting to check whether this corresponds to the expected value or if it transitions to an exponential decay, as mentioned for the short-time dynamics in the supplementary information of Popović et al.

(C) Divergence of Failure Time Near :
The time to failure diverges as . I assume that determining the exponent is challenging due to sample-to-sample fluctuations in for a given annealing degree. Could the authors provide an estimate of these fluctuations in the case of the stable glass?

The second part of the manuscript is more descriptive but highly interesting. From the examples provided, it appears that the plastic deformation map does not allow one to predict where the shear band will form. I find the discussion in its current form appropriate and believe it will be valuable for future studies.

Conclusion:
The manuscript presents a compelling analysis of the creep regime in amorphous materials with varying annealing conditions.

Requested changes

I suggest the authors clarify : 1) the initial strain rate growth 2) further investigate the long-time decay behavior for 3) discuss possible fluctuations in . Addressing these points would strengthen the conclusions and enhance the overall clarity of the study.

Recommendation

Ask for minor revision

  • validity: good
  • significance: high
  • originality: good
  • clarity: high
  • formatting: excellent
  • grammar: perfect

Author:  Pinaki Chaudhuri  on 2025-08-01  [id 5699]

(in reply to Report 1 on 2025-03-29)
Category:
answer to question

We thank the referee for the positive assessment of our work and for the insightful comments. In the attached file, we respond to the received feedback and questions.

Attachment:

response_to_referee.pdf

---

## Round 1 · Referee Report · Anonymous (Referee 2) · 2025-4-29

Strengths

State of the art simulations. Clear exposition of the results. Comparison with a scaling theory.

Report

This paper performs state of the art MD simulations of a simple glass model under shear, prepared at different initial stability through quench afer efficient MCMC at different initial temperature. Fixing the shear stress above the Tini dependent yield threshold the authors study the time scales and physical processes involved in the fluidization of the system, which, differently from the final state, strongly depend on the annealing procedure. The results are compared with recent scaling theories, which appear to work ‘reasonably’ for large initial temperature.

The paper is clear and well written, recommend publication in the present form.

Requested changes

N/A

Recommendation

Publish (easily meets expectations and criteria for this Journal; among top 50%)

  • validity: high
  • significance: high
  • originality: high
  • clarity: top
  • formatting: perfect
  • grammar: perfect

Author:  Pinaki Chaudhuri  on 2025-08-01  [id 5698]

(in reply to Report 2 on 2025-04-29)

We thank the referee for the positive feedback and the recommendation for publication.

---

## Round 2 · Referee Report · Anonymous (Referee 1) · 2025-8-4

Report

The authors answer to my questions and I am happy with the changes and the current version of the manuscript.

Recommendation

Publish (easily meets expectations and criteria for this Journal; among top 50%)

---

## Round 2 · Author Response

We thank the referees for the positive assessment of our work and for the insightful comments. Following up on their feedback, we have revised the text at a few places. The revised version of the manuscript is now being uploaded.

---

## Round 2 · List of Changes

We have made three text changes, following the feedback of Referee #1 --

i) In sub-section III A, we have added the text "Initially, after the shear stress is imposed, $\dot \gamma(t)$ increases linearly with time because we use a first-order barostat in Eq.~(\ref{eq:barostat}) formulated in terms of the strain rate; this early deformation regime is same for all the annealing histories".

ii) In sub-section III A, we have added the text "We observe that after this initial power-law decay, there is a crossover to a faster decay at longer times as the system approached dynamical arrest or jamming. The mechanism behind this ultimate cut-off is still under debate, with proposed explanations including structural aging and the relaxation of residual stresses".

iii) In sub-section III B, we have added the text "We note that determining $\tau_{ss}$ becomes increasingly challenging as the applied stress approaches the threshold stress, because sample-to-sample fluctuations grow dramatically (data not shown). This difficulty is further enhanced for samples with higher initial stability".

---

## Editorial Decision

published